# Influence of Static Navigation Technique on the Accuracy of Autotransplanted Teeth in Surgically Created Sockets

**DOI:** 10.3390/jcm11041012

**Published:** 2022-02-15

**Authors:** Elena Riad Deglow, Nayra Zurima Lazo Torres, David Gutiérrez Muñoz, María Bufalá Pérez, Agustín Galparsoro Catalán, Álvaro Zubizarreta-Macho, Francesc Abella Sans, Sofía Hernández Montero

**Affiliations:** 1Department of Implant Surgery, Faculty of Health Sciences, Alfonso X el Sabio University, 28691 Madrid, Spain; eriaddeg@uax.es (E.R.D.); nlazotor@myuax.com (N.Z.L.T.); dgutierr@uax.es (D.G.M.); mperebuf@uax.es (M.B.P.); agalpcat@uax.es (A.G.C.); shernmon@uax.es (S.H.M.); 2Department of Orthodontics, Faculty of Medicine and Dentistry, University of Salamanca, 37008 Salamanca, Spain; 3Department of Endodontics, Universitat Internacional de Catalunya, 08195 Barcelona, Spain; franabella@uic.es

**Keywords:** accuracy, computed-assisted template, computer-aided static navigation, cone-beam computed tomography scan, digital impression, tooth autotransplantation

## Abstract

The aim of this study was to analyse and compare the position of single-rooted autotransplanted teeth using computer-aided SNT drilling and conventional freehand (FT) drilling, by comparing the planned and performed position at the coronal, apical and angular level. Materials and methods: Forty single-root upper teeth were selected and distributed into the following study groups: A. Autotransplanted tooth using the computer-aided static navigation technique (SNT) (*n* = 20) and B. Autotransplanted tooth using the conventional free-hand technique (FT) (*n* = 20). Afterwards, the teeth were embedded into two experimental models and 10 single-root upper teeth were randomly autotransplanted in each experimental model. The experimental models were submitted to a preoperative cone-beam computed tomography (CBCT) scan and a digital impression by a 3D intraoral scan, in addition to a postoperative CBCT scan, after the autotransplantation. Datasets from postoperative CBCT scans of the two study groups were uploaded to the 3D implant planning software, aligned with the autotransplantation planning, and the coronal, apical and angular deviations were measured. The results were analysed using Student’s *t*-test and Mann–Whitney non-parametric statistical analysis. Results: Coronal (*p* = 0.079) and angular (*p* = 0.208) statistical comparisons did not present statistically significant differences; however, statistically significant differences between the apical deviation of the SNT and FT study groups (*p* = 0.038) were also observed. Conclusions: The computer-aided static navigation technique does not provide higher accuracy in the positioning of single-root autotransplanted teeth compared to the conventional free-hand technique.

## 1. Introduction

Autotransplantation entails transplanting embedded, impacted or erupted teeth from one extraction site to a fresh extraction socket or surgically prepared socket [1]. The advantages of an autotransplanted tooth over a fixed osseointegrated implant include improved resistance to occlusal loading, preservation of the periodontal ligament (PDL) and surrounding bone, continuous bone growth and potentially enhanced aesthetics [2,3,4,5]. Among the indications for tooth autotransplantation are impacted or ectopic teeth, premature and/or traumatic tooth loss, tooth loss resulting from tumours or for iatrogenic reasons, congenitally missing teeth in one arch together with arch length discrepancy, or clinical signs of tooth crowding in the opposing arch, replacement of hopeless teeth and/or developmental dental anomalies [3,6,7].

The challenges of prognosticating root development and dental root resorption post transplantation meant that the 50% success rate of autologous tooth transplantation in the 1950s was relatively low [8,9]. Many studies performed since the 1990s on periodontal tissue and periodontal membrane healing and root resorption have led to a rapid increase in transplant success [3,10,11]. The immature tooth with an open apex is characterized by having an adequate blood supply and stem cells stimulating pulp revascularization post autotransplantation [12]. This revascularization promotes continuous root development and tooth vitality and, at the same time, induces normal alveolar bone growth, which is unfeasible in fixed prostheses. Hence, autotransplantation has a high success rate in immature teeth and is the most conservative and physiologic tooth replacement option [13,14,15,16,17], especially in young patients [6,18].

Some authors have recently determined that there is no significant difference in the success rate of autotransplantation between mature and immature teeth [19,20,21]. In their systematic review, Chung et al. observed that the estimated 1- and 5-year survival rates of autotransplanted teeth with completely formed roots were 98.0% and 90.5%, respectively [6].

The biological responses and wound healing behave similarly to those of avulsed teeth post replantation. Mechanical damage during extraction or continuous traumatic press-fit placement in the recipient socket could harm PDL, giving rise to gradual root resorption. These complications are overcome thanks to improvements in diagnostic and surgical techniques, particularly computer-aided rapid prototyping (CARP) models (tooth replicas) and three-dimensional (3D) printed guiding templates [22,23,24,25]. These digital techniques not only allow clinicians to select the most suitable donor tooth, according to tooth morphology, but also show them the ideal 3D position and the required dimensions of the recipient socket during surgery. Moreover, the use of tooth replicas can reduce extra oral time and possible donor tooth injury during the procedure [22,23].

However, depending on when the tooth was lost, the recipient site conditions may change. In cases of autotransplantation to a fresh extraction socket immediately after extraction of a hopeless tooth, there is usually sufficient bone [10,11,26]. However, for patients with conditions such as congenitally missing teeth or early tooth loss, the recipient site calls for surgical creation [21]. At present, 3D radiologic data are also being used for model-based surgical guides to avoid free-hand preparation of the recipient site [3,16,21,22,23].

Anssari Moin et al. used 10 partially edentulous human mandibular cadavers to assess the accuracy of computer-assisted template-guided autotransplantation with custom 3D designed/printed surgical tools. Their comparison of the superimposed images of the preoperatively planned donor teeth positions and the postoperative donor teeth positions revealed a mean angular deflection (alpha) of 5.6 ± 5.4°. When comparing the bodily 3D positions (a), the authors found a mean deviation of 3.15 ± 1.16 mm, resulting in a mean apical deviation of 2.61 ± 0.78 mm [24]. A comparison of superimposed images of the preoperative planning and the final donor tooth position yielded results similar to those obtained by implant-guided surgery [27]. However, there appears to be no studies comparing precision between free-hand preparation and a static navigation technique (SNT) using an implant drilling sequence.

The aim of the present study was to analyse and compare the position of single-rooted autotransplanted teeth using computer-aided SNT drilling and conventional freehand (FT) drilling, by comparing the planned and performed position at the coronal, apical and angular levels. The null hypothesis (H0) was that there is no difference between computer-aided SNT and conventional FT concerning the accuracy of single-rooted autotransplanted teeth.

## 2. Materials and Methods

### 2.1. Study Design

Forty single-rooted maxillary anterior teeth (incisors and canines), extracted for periodontal or orthodontic reasons, were selected for this study conducted at the Dental Centre of Innovation and Advanced Specialties at Alfonso X El Sabio University (Madrid, Spain) between March and April 2021. The sample size was selected according to a previous study with a power effect of 88.4 (it is considered acceptable from 80) [28]. The manuscript of this laboratory study has been written according to 2021 Preferred Reporting Items for Laboratory studies in Endodontology (PRILE) guidelines (Figure 1) [29,30]. In addition, the study was conducted in accordance with the principles defined in the German Ethics Committee’s statement for the use of organic tissues in medical research (Zentrale Ethikkommission, 2003) and was authorized by the Ethical Committee of the Faculty of Health Sciences, University Alfonso X el Sabio (Madrid, Spain), in October 2020 (Process No. 05/2020). All the patients signed an informed consent form to donate the teeth for the present study.

### 2.2. Experimental Procedure

The single-rooted teeth were embedded into two experimental epoxy resin models (Ref. 20-8130-128, EpoxiCure^®^, Buehler, IL, USA), each with 20 teeth. Ten teeth (for autotransplantation) were placed in the internal part of the model, and 10 teeth (used as a reference), in the external part. The teeth were randomly (Epidat 4.1, Galicia, Spain) assigned to two study groups: Group A, autotransplanted teeth using a computer-aided static navigation technique (NemoScan^®^, Nemotec, Madrid, Spain) (SNT) (*n* = 20), and Group B, autotransplanted teeth using conventional free-hand technique (FT) (*n* = 20).

The two experimental models were submitted to a preoperative cone-beam computed tomography (CBCT) scan (WhiteFox, Acteón Médico-Dental Ibérica S.A.U.-Satelec, Merignac, France) with the following exposure parameters: 105.0 kilovolt peak, 8.0 milliamperes, 7.20 s, and a field of view of 15 × 13 mm (Figure 2A). Subsequently, a digital impression was made using a 3D intraoral scan (True Definition, 3M ESPE™, Saint Paul, MN, USA) by means of 3D in-motion video imaging technology to generate a standard tessellation language (STL) digital file (Figure 2B). The 3D intraoral scan (True Definition) uses a cloud of points that create a tessella network, representing 3D objects as polygons composed of equilateral triangle tessellas [31,32]. The image capture procedure was performed by scanning the palatine and occlusal surface followed by the buccal surface, according to the manufacturer’s recommendations. Datasets obtained from this digital workflow were uploaded to a 3D implant planning software (NemoScan^®^) to plan the placement of autotransplantation in Group A (Figure 2C).

After matching the 3D surface scan and CBCT data (WhiteFox), each tooth in the internal part of the model was individually segmented and virtually placed between the teeth placed outside of the model (Figure 3).

The surgically created sockets of the teeth were randomly assigned to the SNT study group; the drilling was performed by means of a 3D printed tooth-supported surgical template with 10 drilling sleeves of 2.5 mm in diameter (NemoScan^®^) (Figure 4). The dimensions of the osteotomy site preparations were designed virtually by superimposing the virtual surgical drilling burs (BioHorizons; Birmingham, AL, USA) on the roots of the autotransplanted teeth. A surgical template was then exported as an STL digital file and 3D printed for fabrication (Explora 3D Lab, Nemotec S.L, Arroyomolinos, Madrid, Spain) with medical-use resin. The osteotomy site was manually drilled with surgical burs according to each root anatomy (BioHorizons).

On the other hand, the drilling procedure of the osteotomy site of the teeth randomly assigned to the autotransplanted tooth using conventional FT study group was performed completely manually. Subsequently, the teeth placed inside of the experimental models of epoxy resin were extracted and placed between the teeth placed outside of the experimental model until it adjusted to the previously autotransplanted planned position (Figure 4). A single operator with 10 years of surgical experience performed all autotransplanted teeth procedures.

### 2.3. Measurement Procedure

After performing the osteotomy site preparation and placing the autotransplanted teeth of both study groups, a postoperative CBCT scan (WhiteFox) of the experimental models were taken with the same, previously described exposure parameters. STL digital files from the planning and datasets from postoperative CBCT scans of the two study groups were uploaded to the 3D implant planning software (NemoScan^®^) and aligned using the 3D implant planning software (NemoScan^®^) to analyse the deviation angle (measured in the centre of the cylinder) and horizontal deviation (measured at the coronal entry point and apical endpoint) (Figure 5) by an independent observer.

### 2.4. Statistical Tests

All the variables of interest were recorded for statistical analysis with SAS v9.4 (SAS Institute Inc., Cary, NC, USA). Descriptive statistical analysis was expressed as means and standard deviations (SDs) for quantitative variables. Comparative analysis was performed by comparing the mean deviation between planned and performed autotransplanted tooth using Student’s *t*-test, since variables had normal distribution, or Mann–Whitney non-parametric test; *p* < 0.05 was considered statistically significant.

## 3. Results

The means and standard deviation (SD) values for coronal, apical and angular deviation of the autotransplanted tooth using computer-aided static navigation technique and conventional freehand technique are displayed in Table 1.

Mean comparison of the coronal deviation of the autotransplanted teeth randomly assigned to the SNT study group did not show a normal distribution; therefore, the comparative analysis was performed by a Mann–Whitney non-parametric test. Median comparison of the autotransplanted teeth revealed no statistically significant differences at the coronal deviation (*p* = 0.079) between the SNT (5.40 ± 3.76 mm) and FT (4.20 ± 1.85 mm) study groups (Figure 6).

Mean comparison of the apical deviation of the autotransplanted teeth randomly assigned to the SNT study group did not show a normal distribution; therefore, the comparative analysis was performed again with a Mann–Whitney non-parametric test. Median comparison of the autotransplanted teeth revealed statistically significant differences at the apical deviation (*p* = 0.038) between SNT (5.65 ± 2.81 mm) and FT (3.90 ± 1.99 mm) study groups (Figure 7).

A mean comparison of the angular deviation of the autotransplanted teeth randomly assigned to the SNT study group showed a normal distribution; therefore, the comparative analysis was performed using Student’s *t*-test. Mean comparison of the autotransplanted teeth revealed no statistically significant differences at the angular deviation (*p* = 0.208) between SNT (5.65 ± 2.81 mm) and FT (3.90 ± 1.99 mm) study groups (Figure 8).

## 4. Discussion

The present study reported that the coronal and angular deviations between the SNT and FT study groups did not show statistically significant differences; however, statistically significant differences were observed between the apical deviation of the SNT and FT study groups. One of the main problems in dentistry is the premature loss of teeth resulting from trauma, caries or malformations, especially in growing patients [33]. With this in mind, the clinician can choose from various treatment options depending on the patient’s age [23]. The most common restorative approaches for adults include fixed or removable partial dentures, implants or orthodontics. However, in paediatric and adolescent patients, implant placement is totally contraindicated [34]. Accordingly, in determined patients, autologous transplantation offers an effective treatment option with the potential to restore masticatory function and aesthetics [10]. Unlike implants, transplanted teeth behave in the same way as any natural tooth, both of which maintain the alveolar bone and occlusion during growth. The benefit of this procedure is that it allows the replacement of a hopeless or missing tooth with another tooth from the same patient [34].

Depending on the time of autotransplantation, the technique can be performed in either fresh extraction sockets or surgically created sockets [33]. In an immediate autotransplantation, fibroblasts and PDL remaining in the socket wall proliferate and migrate to the blood clot, promoting bone and connective tissue reconstruction and significantly aiding the revascularization of the root surface of the donor tooth [35]. One potential drawback to immediate autotransplantation is that the donor tooth may not fit perfectly into the recipient socket, which results in a discrepancy between the tooth surface and the alveolar wall. For bone formation to take place, it is essential for the root surface of the donor tooth to be near the cervical level of the adjacent bone, since the underlying tissue acts like a closed wound, reducing the possibility of infection and complications [2]. There are some indications for late autotransplantation placement according to patient- or site-specific reasons. These include patients with congenitally missing teeth or premature tooth loss, or when there is insufficient mesio-distal space in the recipient area, for which subsequent orthodontic treatment is needed [33]. Although this technique is more challenging, no significant difference in outcomes compared to autotransplantation in fresh extraction sockets have been observed [36,37]. After tooth extraction, the buccal and lingual walls of the alveolus resorb significantly [38]. In this situation, the root of the donor tooth can be rotated or even resected to fit within the new socket. The clinician may choose from surgical drills, implant drills or even trephines to surgically create the new socket [21]. The main factors determining a successful autotransplantation involve preserving the PDL and correctly adapting tissue [39]. Hence, it is crucial to avoid excessive manipulation of the tooth and minimize both the extra-alveolar time (should not exceed 12 min) and the distance between the alveolus and the root of the tooth [24,25,34]. This is particularly relevant in surgically created sockets, in which revascularization is delayed, leading to insufficient nutrition of the apical tissues, negatively affecting the vitality of Hertwig’s epithelial root sheath (HERS) [40,41]. This is a highly technical and sensitive surgical procedure that demands all the clinician’s experience and skill [25]. In the conventional autotransplantation technique, whose first clinical application dates to 1950 [41], the donor tooth served as a template to prepare the socket, which involved excessive manipulation of the donor tooth, greater chemical and physical trauma to the PDL and more extra-alveolar time. However, with the advent of CBCT and digital planning, the complexity and failure rate of this technique has been substantially reduced [4,18,22]. In 2001, Lee et al. described the use of computer-assisted replicas of the donor tooth, making it possible to prepare the recipient socket without having to use the donor tooth itself [22]. In addition, available surgical planning software allows the clinician to design and manufacture 3D-printed surgical guides. These guides approximate autotransplantation surgery to guided surgery, but the literature analysed has shown that certain inaccuracies between the original digitally planned position and the final position of the donor tooth remain frequent. For a guided surgery to be closer to its original digital planning, it is essential to achieve a precise osteotomy that produces minimal trauma to the recipient area [4]. Trauma is directly related to overheating of the bone during osteotomy, which can lead to cell death, preventing new bone formation. In addition, an overlarge alveoloplasty increases the discrepancy between the donor tooth and the recipient area, causing instability of the blood clot and impeding periodontal regeneration [1,7]. Given the limited studies in which personalized, and 3D-designed surgical appliances are used in guided autotransplantation, the authors of the present study evaluated the accuracy and success of this surgical approach. Anssari Moin et al. in their study using guided autotransplantation with surgical splints and customized surgical instruments on human mandibular cadaver jaws reported a mean coronal deviation of 3.15 +/− 1.16 mm, a mean apical deviation of 2.61 +/− 0.78 mm and a mean angular deviation of 5.6–5.4° between the digitally planned position and the definitive position of the transplanted teeth [24]. These values are within the generally accepted ranges for surgical guides in implant treatment; however, they may be clinically relevant for autotransplantation, and thus these results should be improved. Moreover, Wu et al. also reported a high accuracy of the static surgical guides for dental implant placement [42]. Impacted teeth are often considered candidates for tooth autotransplantation and Cavuoti et al. highlighted the risk of root resorption in impacted teeth and recommended repositioning the impacted tooth to prevent root resorption [43]. However, our results are difficult to compare with those of Anssari Moin et al., who expressed the results as means and standard deviation [24]. Due to the fact that the mean comparison of both the coronal and apical deviations of the autotransplanted teeth in our study showed no normal distribution, a statistical analysis was performed by median comparison using a Mann–Whitney non-parametric test. The inherent deficiencies in the digital workflow were related to the lack of precision regarding the result. Ender et al. reported less statistical accuracy (*p* < 0.05) of the digital impressions of the partial-arch than of the digital impressions of the total-arch [44]. In the present study, the imprecision of the manual segmentation may have influenced the choice of bur size, which may have resulted in inadequate drilling. In addition, the sockets were surgically created using a single bur, but the oval section of the autotransplanted teeth required additional manual drilling to adapt the root anatomy of the donor tooth to the socket, which may have influenced the definitive position. In addition, it is essential to analyse the root morphology to avoid fractures during dental extraction manoeuvres and complications during the autotransplantation procedure (especially in dislacerated roots or divergent roots of multirooted teeth). Likewise, it is necessary to evaluate the mesio-distal size of the edentulous space to be rehabilitated by the tooth to be autotransplanted and the occlusal contacts. The authors recommend evaluating these parameters in the surgical planning phase. Moreover, the trueness of the intraoral scanner has been highlighted as a relevant factor, since it can induce the appearance of a clinically relevant cumulative error; however, the present study used powder-dependent intraoral scanners, which are significantly better (*p* < 0.05) than non-powder-dependent scanners as the translucency they produce shows fewer errors in the images [45].

Finally, the authors of the present study suggest there is a need to conduct a study on cryopreserved cadavers to evaluate the precision and reproducibility of the technique using different splints and customized surgical instruments, given the dearth of studies on guided surgery to create neo-alveolus in autotransplantations. Furthermore, the deviation between the planned and final position of the surgically guided autotransplantation in this study should also be assessed. Additionally, the experimental nature of the study allows for better 3D visibility and perception compared to a clinical situation.

## 5. Conclusions

Within the limitations of this in vitro study, the results show that the computer aided SNT was less reliable than FT and the use of SNT in the clinic should be suspended until further research is conducted. Specifically, coronal and angular deviations between the computer aided SNT and FT study groups did not show statistically significant differences; however, statistically significant differences were observed between the apical deviation of the SNT and FT study groups.

## Figures and Tables

**Figure 1 jcm-11-01012-f001:**
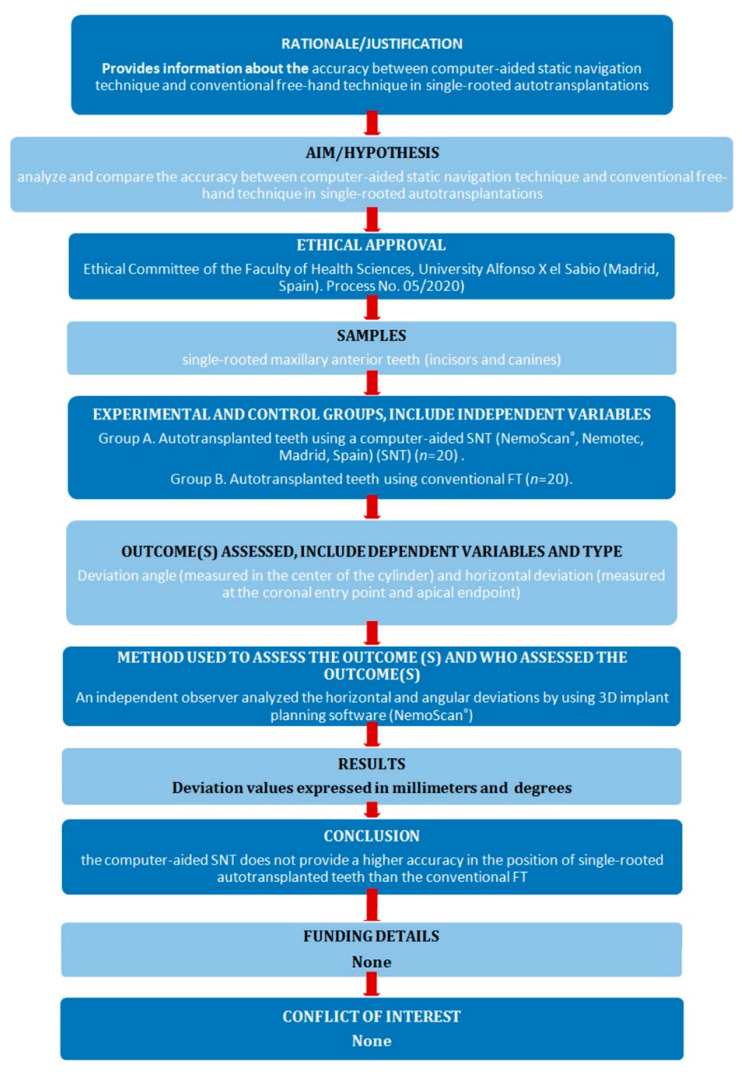
Preferred reporting items for laboratory studies in endodontology flowchart.

**Figure 2 jcm-11-01012-f002:**
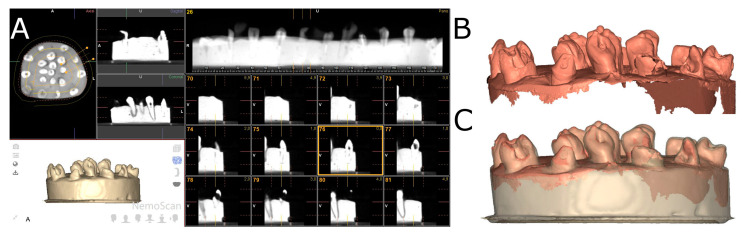
(**A**) CBCT scan, (**B**) STL digital files and (**C**) alignment of the digital workflow.

**Figure 3 jcm-11-01012-f003:**
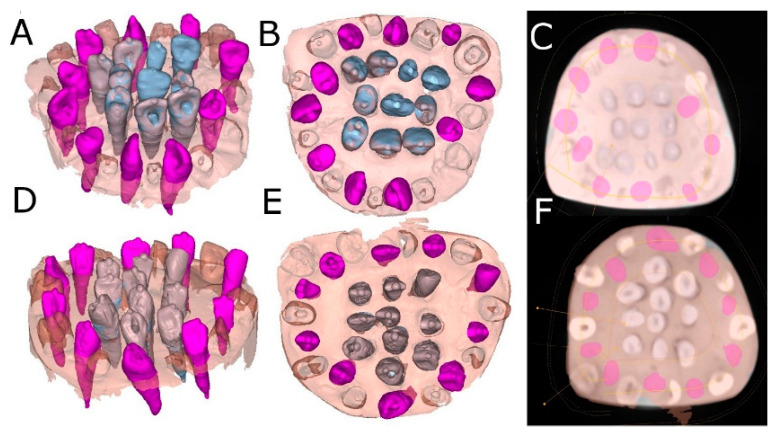
(**A**) Frontal, (**B**) occlusal and (**C**) apical view of the individually segmented (blue) and autotransplanted teeth (purple) between the teeth placed outside of the experimental model (pink) randomly assigned to the SNT study group. (**D**) Frontal, (**E**) occlusal and (**F**) apical view of the individually segmented (grey) and autotransplanted teeth (purple) between the teeth placed outside of the experimental model (pink) randomly assigned to the FT study group.

**Figure 4 jcm-11-01012-f004:**
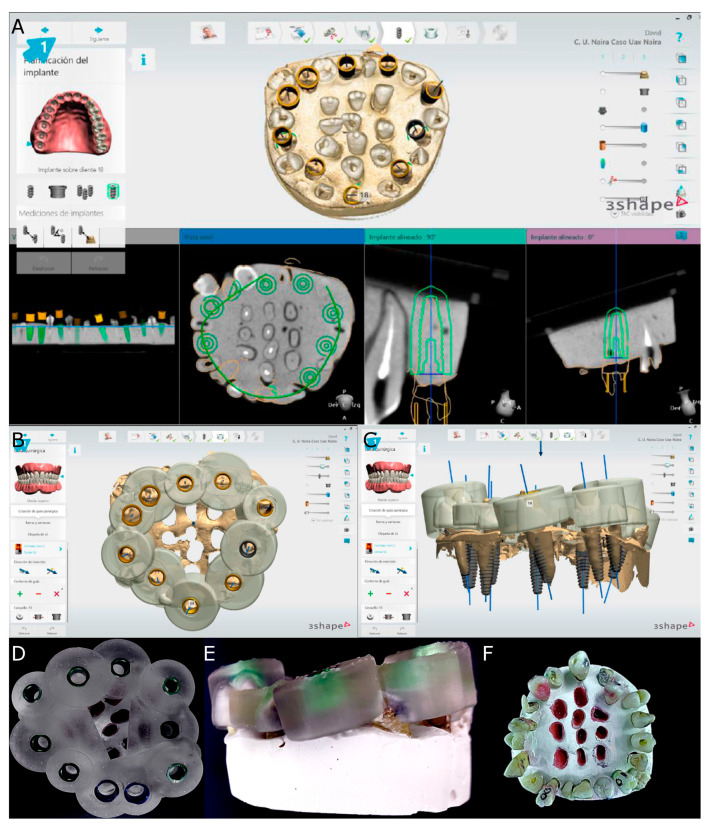
(**A**) Drilling site planning in the CBCT scan, (**B**) occlusal view of the surgical template design for computer-aided static navigation technique, (**C**) lateral view of the drilling bur selection according to the root dimensions, (**D**) occlusal and (**E**) lateral view of the surgical template manufactured and (**F**) occlusal view of the transplanted teeth.

**Figure 5 jcm-11-01012-f005:**
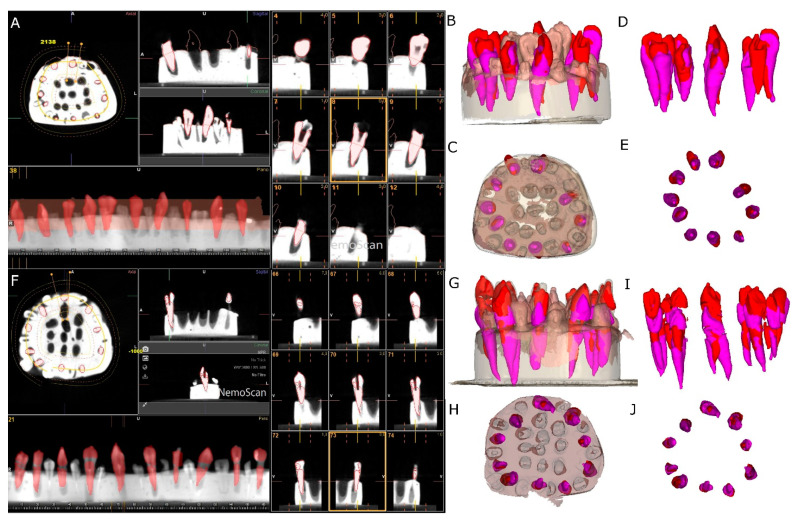
(**A**) Segmented teeth on the postoperative CBCT scan (red teeth), (**B**) lateral view and (**C**) apical view of the experimental models and (**D**) lateral and (**E**) apical view of the planned (pink teeth) and performed (red teeth) autotransplanted teeth without model of the conventional freehand technique study group. (**F**) Segmented teeth on the postoperative CBCT scan (red teeth), (**G**) lateral and (**H**) apical view of the experimental models and (**I**) lateral and (**J**) apical view of the planned (pink teeth) and performed (red teeth) autotransplanted teeth without model of the computer-aided static navigation technique.

**Figure 6 jcm-11-01012-f006:**
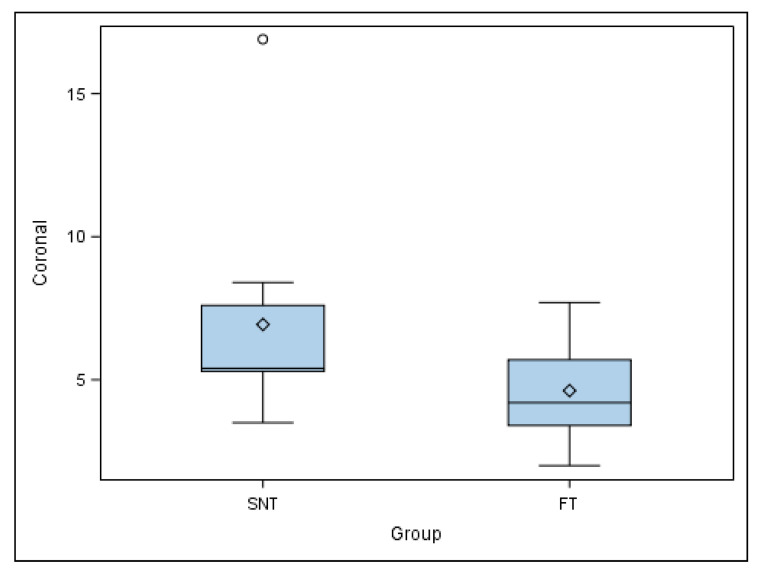
Box plot of the coronal deviation of the autotransplanted teeth. The horizontal line in each box represents the respective median value of the study groups. ◊: Mean value of the box plots. ◦: Means and extreme value.

**Figure 7 jcm-11-01012-f007:**
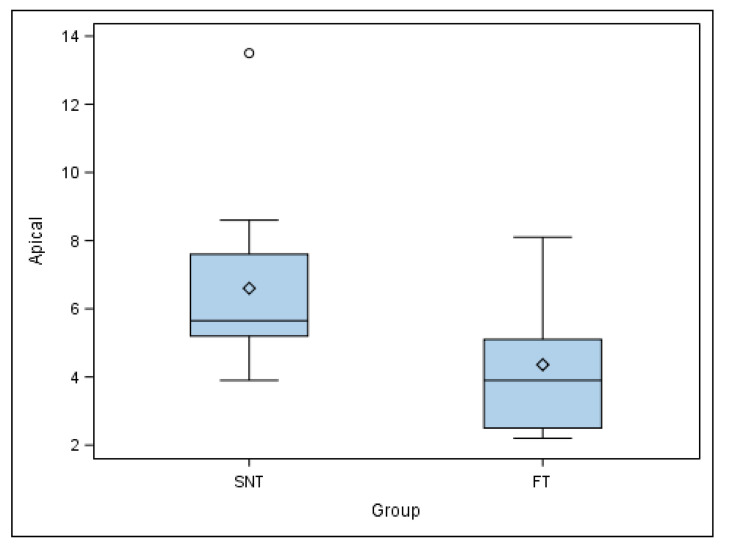
Box plot of the apical deviation of the autotransplanted teeth. The horizontal line in each box represents the respective median value of the study groups. ◊: Mean value of the box plots. ◦: Means and extreme value.

**Figure 8 jcm-11-01012-f008:**
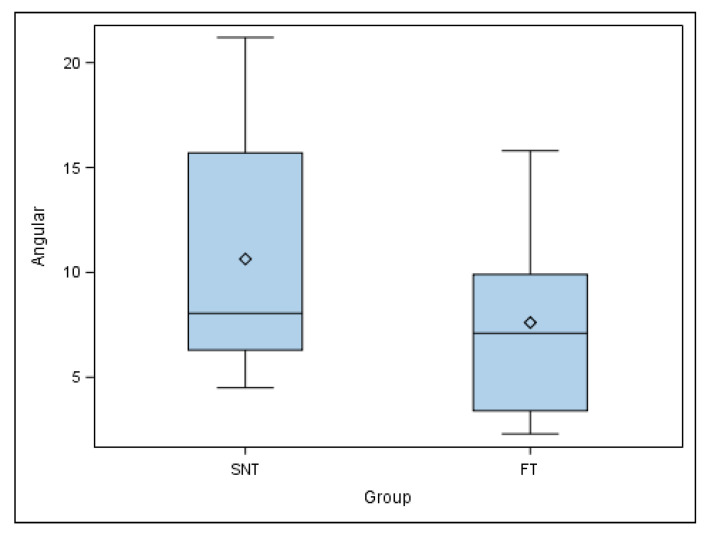
Box plot of the angular deviation of the autotransplanted teeth. The horizontal line in each box represents the respective median value of the study groups. ◊: Mean value of the box plots.

**Table 1 jcm-11-01012-t001:** Descriptive deviation values at coronal (mm), apical (mm) and angular (°) levels of the autotransplanted tooth using computer-aided static navigation technique and conventional free-hand technique.

		*n*	Mean	Median	SD	Minimum	Maximum
SNT	Coronal	10	6.93	5.40 ^a^	3.76	3.50	16.90
	Apical	10	6.60	5.65 ^a^	2.81	3.90	13.50
	Angular	10	10.64	8.05 ^a^	5.78	4.50	21.20
FT	Coronal	10	4.62	4.20 ^a^	1.85	2.00	7.70
	Apical	10	4.36	3.90 ^b^	1.99	2.20	8.10
	Angular	10	7.61	7.10 ^a^	4.53	2.30	15.80

SNT: static navigation technique. FT: free-hand technique. ^a,b^ Statistically significant differences between groups (*p* < 0.05).

## Data Availability

Data available on request due to restrictions, e.g., privacy or ethical.

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
