# Peer review of "Influence of Static Navigation Technique on the Accuracy of Autotransplanted Teeth in Surgically Created Sockets"

_jcm, 2022, doi:10.3390/jcm11041012_

Round 1

Reviewer 1 Report

This study was to investigate the accuracy between computer-aided static navigation technique and conventional free-hand technique in single-root autotransplantation. The topic is interesting. but I have some comments:

  • Please explain how to calculate the sample size.
  • Please provide high-quality images.
  • Please give the limation and outlook this study
  • Is there any standardized protocol for this experimental model?

Author Response

Dear Reviewer 1,

I’m pleased to resubmit the manuscript of the work entitled, “Influence of Static Navigation Technique on the Accuracy of Autotransplanted Teeth in surgically created sockets”

Reviewer 1: English language and style are fine/minor spell check required

Response: In order to adapt to the reviewer's 1 comments, we have sent the manuscript to the English Editing Service of MDPI. We attached the Certificate.

Reviewer 1: Please explain how to calculate the sample size.

Response: In order to adapt to the reviewer's 1 comments, we clarify that we used the results obtained in a previous study (Zubizarreta-Macho Á, Muñoz AP, Deglow ER, Agustín-Panadero R, Álvarez JM. Accuracy of Computer-Aided Dynamic Navigation Compared to Computer-Aided Static Procedure for Endodontic Access Cavities: An in Vitro Study. J Clin Med. 2020, 9, 129. doi: 10.3390/jcm9010129) to calculate sample size. We added this information at the “Study Design” section.

Reviewer 1: Please provide high-quality images

Response: In order to adapt to the reviewer's 1 comments, we have changed Figure 1 for another of higher quality.

Reviewer 1: Please give the limation and outlook this study

Response: In order to adapt to the reviewer's 1 comments, we clarify that the experimental nature of the study allows for better three-dimensional visibility and perception compared to a clinical situation.

Reviewer 1: Is there any standardized protocol for this experimental model?

Response: In order to adapt to the reviewer's 1 comments, we clarify that a surgical protocol has been previously described (Anssari Moin D, Verweij JP, Waars H, van Merkesteyn R, Wismeijer D. Accuracy of computer-assisted template-guided autotransplantation of teeth with custom three-dimensional designed/printed surgical tooling: a cadaveric study. J Oral Maxillofac Surg. 2017, 75, 925.e1-925.e7. doi: 10.1016/j.joms.2016.12.049.) to auto-transplant teeth using surgical splints; however, is necessary to assess the accuracy of this surgical procedure through experimental studies, before applying it to clinical practice with patients.

We take this opportunity to thank the recommendations and suggestions made by the reviewers to improve the document. 

Yours sincerely,

Reviewer 2 Report

The research is well designed and carried out.

Abstract: it is a good summary of the paper, and it it well organized.

Introduction contains enough background informations regarding the techniques involved and adequate references.

Materials and methods are clearly described, but Figure 1 has a very low resolution, please improve it.

Results: statistical analysis is carried out in a proper way, but the samples involved few participants. I appreciate that you state it as a drawback of your work in discussion. Please add some other clinical consideration in discussions.

Conclusions must be improved.

Author Response

Dear Reviewer 2,

I’m pleased to resubmit the manuscript of the work entitled, “Influence of Static Navigation Technique on the Accuracy of Autotransplanted Teeth in surgically created sockets”

Reviewer 2: English language and style are fine/minor spell check required

Response: In order to adapt to the reviewer's 2 comments, we have send the manuscript to the English Editing Service of MDPI. We attached the Certificate.

Reviewer 2: Materials and methods are clearly described, but Figure 1 has a very low resolution, please improve it.

Response: In order to adapt to the reviewer's 2 comments, we have changed Figure 1 for another of higher quality.

Reviewer 2: Please add some other clinical consideration in discussions

Response: In order to adapt to the reviewer's 2 comments, we have added more clinical considerations in the Discussion section.

Reviewer 2: Conclusions must be improved.

Response: In order to adapt to the reviewer's 2 comments, we have added that “Specifically, coronal and angular deviations between the computer-aided SNT and FT study groups did not show statistically significant differences; however, statistically significant differences were observed between the apical deviation of the SNT and FT study groups”, in the “Conclusion” section.

We take this opportunity to thank the recommendations and suggestions made by the reviewers to improve the document.

Yours sincerely,

Reviewer 3 Report

In the manuscript entitled: “Influence of Static Navigation Technique on the Accuracy of Autotransplanted Teeth in surgically created sockets”, the authors compared the accuracy between computer-aided static navigation technique and conventional free-hand technique in single-root autotrasplantation.

The authors found that the coronal (p = 0.079) and angular (p = 0.208) statistical comparisons did not presented statistically significant differences; however, statistically significant differences between the apical deviation of the SNT and FT study groups (p = 0.038) were also observed.

The authors concluded that the computer-aided static navigation technique does not provide a higher accuracy on the positioning of single-root autotransplanted teeth comparing to conventional free-hand technique.

Major comments:

In general, the idea and innovation of this study, regards analysis of SNT in oral surgery is interesting, because the role these aspects in dentistry are validated but further studies on this topic could be an innovative issue in this field could be open a creative matter of debate in literature by adding new information. Moreover, there are few reports in the literature that studied this interesting topic with this kind of study design.

The study was well conducted by the authors; However, there are some concerns to revise that are described below.

The introduction section resumes the existing knowledge regarding the important factor linked with challenges of prognosticating root development and dental root resorption.

However, as the importance of the topic, the reviewer strongly recommends, before a further re-evaluation of the manuscript, to update the literature through read, discuss and must cites in the references with great attention all of those recent interesting articles, that helps the authors to better introduce and discuss the role of SNT in root resorption of impacted canines or third molar: 1) Angle Orthod. 2016 Jul;86(4):681-91. doi: 10.2319/050615-309.1. 2) Int J Implant Dent. 2020 Nov 24;6(1):78.

The authors should be better specified, at the end of the introduction section, the rational of the study and the aim of the study. Moreover, specify the number of clinicians that were involved in the different stages of the study.

The discussion section appears well organized with the relevant paper that support the conclusions, even if the authors should better discuss the exactly role of periodontal biotype on Static Navigation Technique. The conclusion should reinforce in light of the discussions.

In conclusion, I am sure that the authors are fine clinicians who achieve very nice results with their adopted protocol. However, this study, in my view does not in its current form satisfy a very high scientific requirement for publication in this journal and requests a revision before a futher re-evaluation of the manuscript.

Minor Comments:

Abstract:

  • Better formulate the abstract section by better describing the aim of the study

Introduction:

  • Please refer to major comments

Discussion

  • Please add a specific sentence that clarifies the results obtained in the first part of the discussion
  • Page 11 last paragraph: Please reorganize this paragraph that is not clear

Author Response

Dear Reviewer 3,

I’m pleased to resubmit the manuscript of the work entitled, “Influence of Static Navigation Technique on the Accuracy of Autotransplanted Teeth in surgically created sockets”

Reviewer 3: English language and style are fine/minor spell check required

Response: In order to adapt to the reviewer's 3 comments, we have send the manuscript to the English Editing Service of MDPI. We attached the Certificate.

Reviewer 3: the reviewer strongly recommends, before a further re-evaluation of the manuscript, to update the literature through read, discuss and must cites in the references with great attention all of those recent interesting articles, that helps the authors to better introduce and discuss the role of SNT in root resorption of impacted canines or third molar: 1) Angle Orthod. 2016 Jul;86(4):681-91. doi: 10.2319/050615-309.1. 2) Int J Implant Dent. 2020 Nov 24;6(1):78.

Reviewer 3: The authors should be better specified, at the end of the introduction section, the rational of the study and the aim of the study.

Reviewer 3: Moreover, specify the number of clinicians that were involved in the different stages of the study.

Reviewer 3: the authors should better discuss the exactly role of periodontal biotype on Static Navigation Technique. The conclusion should reinforce in light of the discussions

Reviewer 3: Abstract: Better formulate the abstract section by better describing the aim of the study

Reviewer 3: Discussion: Please add a specific sentence that clarifies the results obtained in the first part of the discussion

Reviewer 3: Discussion: Page 11 last paragraph: Please reorganize this paragraph that is not clear

We take this opportunity to thank the recommendations and suggestions made by the reviewers to improve the document.

Yours sincerely,

Round 2

Reviewer 1 Report

The manuscript can be considered for publication.

Reviewer 3 Report

The authors have well addressed to all comments raised by the reviewer. No further issues are needed in the present version of the manuscript.